# Development of a multiphase chemical mechanism to improve secondary organic aerosol formation in CAABA/MECCA (version 4.5.6-rc.1)

Felix Wieser[1], Rolf Sander[2], Changmin Cho[1,a], Anna Novelli[1], Hendrik Fuchs[3], Ralf Tillmann[1], Thorsten Hohaus[1], and Domenico Taraborrelli[1]

[1]Forschungszentrum Jülich GmbH, Institute of Energy and Climate Research, IEK-8: Troposphere, Jülich, Germany
[2]Atmospheric Chemistry Department, Max Planck Institute for Chemistry, Mainz, Germany
[3]Department of Physics, University of Cologne, Cologne, Germany
[a]Now at: Atmospheric Chemistry Observations and Modeling Laboratory, National Center for Atmospheric Research, Boulder, USA

**Correspondence:** Felix Wieser (f.wieser@fz-juelich.de) and Domenico Taraborrelli (d.taraborrelli@fz-juelich.de)

**Abstract.** During the last decades, the impact of multiphase chemistry on secondary organic aerosol (SOA) has been demonstrated to be key in explaining laboratory experiments and field measurements. However, global atmospheric models still show large biases when simulating atmospheric observations of organic aerosols (OA). Major reasons for the model errors are the use of simplified chemistry schemes of gas-phase oxidation of vapors and parameterization of heterogeneous surface reactions. The photochemical oxidation of anthropogenic and biogenic volatile organic compounds (VOCs) leads to products that either produce new SOA or are taken up by existing aqueous media like cloud droplets and deliquescent aerosols. After partitioning, aqueous-phase processing results in polyols, organosulfates, and other products with a high molar mass and oxygen content. In this work, we have introduced the formation of new low-volatility organic compounds (LVOCs) into the multiphase chemistry box model CAABA/MECCA. Most notable are the additions of the SOA precursors limonene, n-alkanes (5 to 8 C atoms), and a semi-explicit chemical mechanism for the formation of LVOCs from isoprene oxidation in the gas- and aqueous-phase. Moreover, Henry's law solubility constants and their temperature dependences have been estimated for the partitioning of organic molecules to the aqueous-phase. Box model simulations indicate that the new chemical scheme predicts enhanced formation of LVOCs, which are accounted for being precursor species to SOA. As expected, the model predicts that LVOCs are positively correlated to temperature but negatively correlated to $NO_x$ levels. However, the aqueous-phase processing of isoprene-epoxydiols (IEPOX) displays a more complex dependence on these two key variables. Semi-quantitative comparison with observations from the SOAS campaign suggests that the model may overestimate methylbutane-1,2,3,4-tetrol (MeBuTETROL) from IEPOX. Further application of the mechanism in the modeling of two chamber experiments (limonene-ozone and isoprene-$NO_3$) shows a sufficient agreement to experimental results within model limitations. The extensions in CAABA/MECCA will be transferred to the 3D-atmospheric model MESSy for a comprehensive evaluation of the impact of aqueous-/aerosol-phase chemistry on SOA at a global scale in a follow up study.

# 1   Introduction

Secondary organic aerosol (SOA) is formed from anthropogenic and biogenic volatile organic compounds (VOCs) in the atmosphere and comprises a significant amount of the total organic aerosol (OA) mass (Hallquist et al., 2009). Atmospheric aerosols have received increased attention in recent years, as a result of their impact on human health, urban visibility, and climate change (Zhong and Jang, 2011; Zhu et al., 2017; Lin et al., 2017; Chen et al., 2020). New SOA precursors, formation pathways, and loss reactions in the gas- and the aqueous-phase have been discovered (Lim et al., 2010; Hodzic et al., 2014; Wennberg et al., 2018). In current models, aqueous-phase pathways leading to SOA are continuously improved, as their importance has been demonstrated in experiments (Carlton et al., 2007; Lin et al., 2014; Ervens, 2015). Hu et al. (2015) investigated aqueous SOA (aqSOA) tracers from isoprene oxidation and found a contribution between 6 % and 36 % to total OA.

Recent models show different biases in their results. Traditional global models generally underestimate SOA mass, especially during haze events, due to the increase in pollutant concentrations (Heald et al., 2011; Tilmes et al., 2019). This gap between models and observations is continuously decreasing in recent years, but SOA formed in available aqueous-phase is often still neglected (Shrivastava et al., 2017). Aqueous SOA is known to be more oxidized than SOA formed from gas-phase precursors, thus the modeled O/C ratios commonly do not match experimental results (Lim et al., 2010). Additionally, products from aqueous-phase oxidation are formed on different time-scales than from gas-phase oxidation. Combined with aqueous- and gas-phase properties (pollutants, pH), this impacts the vertical distribution of SOA in the atmosphere depending on the simulation setup (Seinfeld and Pandis, 2016; Hodzic et al., 2016).

We have implemented recent experimental results and formerly neglected SOA precursors into the chemistry scheme of the box model CAABA/MECCA. The updated chemical mechanism is meant to be used for further advancing global simulations of SOA with the EMAC model (Pozzer et al., 2022). EMAC is the global configuration of the Modular Earth Submodel System (MESSy) (Jöckel et al., 2006; Tost et al., 2006; Kerkweg et al., 2007; Jöckel et al., 2010, 2016). This approach has two advantages: 1) model simulations in the test phase are considerably faster in the box than in the global model; 2) thanks to the MESSy interface structure, the MECCA chemical mechanism can directly be used in the global model EMAC. This allows us to be fast in the implementation process while being able to use the new chemistry in the global model without further adaptations. The partitioning to the aqueous-/particle-phase plays an important role in the SOA formation process. As temperature-dependent Henry's law solubility constants ($H_{s}$) for large organic molecules are sparse in the literature, these properties have to be estimated. The corresponding approach is discussed in Sect. 2.4. In Section 3 the new chemical scheme is evaluate against model, observational, and chamber data. In the following section, we introduce major specifications of the CAABA/MECCA model.

## 2 Model description

### 2.1 Specifications

CAABA/MECCA is a combination of the box model CAABA (Chemistry As A Box model Application) and the atmospheric chemistry model MECCA (Module Efficiently Calculating the Chemistry of the Atmosphere) (Sander et al., 2019). The chemistry module MECCA is also used in the global model EMAC, simplifying the adoption of box model changes into the global model. MECCA contains both gas- and aqueous-phase chemistry and the species can partition between the phases. Simulations with solely gas-phase are also possible to simulate dry conditions. For the partitioning, Henry's law solubility constants are utilized. Non-linear effects like "salting in" and "salting out" influence Henry's law solubility constants. It has been shown that the partitioning of compounds can be substantially influenced by the aerosol salt concentration (Kampf et al., 2013; Herrmann et al., 2015). Similarly, aqueous-phase reaction rates can be influenced by salt composition (Mekic and Gligorovski, 2021). The model does not account for these effects, due to the insufficient data availability (Sander, 2023). This may result in partially incorrect partitioning rates.

Available mechanisms for organic chemistry are the Mainz Organic Mechanism (MOM) (Sander et al., 2019) and the Jülich Aqueous-phase Mechanism of Organic Chemistry (JAMOC) by Rosanka et al. (2021). MOM is the default oxidation mechanism of VOCs in MECCA. It contains an advanced treatment of chemistry of isoprene (Taraborrelli et al., 2012; Nölscher et al., 2014; Novelli et al., 2020), monoterpenes (Hens et al., 2014; Mallik et al., 2018) and aromatics (Cabrera-Perez et al., 2016; Taraborrelli et al., 2021). JAMOC is a reduced subset of the CLEPS mechanism (Mouchel-Vallon et al., 2017). Although built for global chemistry simulations, both chemical mechanisms together cover a wide range of reactions. Combined, they include gas-phase oxidation and gas/aqueous partitioning for species up to 11 carbons and aqueous-phase oxidation for species up to four carbons. The submodel JVAL calculates photochemical rate constants ($j$ values) based on cross-sections and quantum yields (Sander et al., 2014). The related $j$ values are usually first determined for the smallest compound of a compound class and this value is applied to higher homologs. For instance, $j$ values for the photolysis of all organic hydroperoxides are taken equal to the one for methyl hydroperoxide ($j_{\mathrm{CH_3OOH}}$). This is also done in the Master Chemical Mechanism (MCM) (Jenkin et al., 1997). More details about CAABA/MECCA can be found elsewhere (Sander et al., 2005, 2011, 2019).

### 2.2 Model limitations

The main goal of the CAABA/MECCA box model is the investigation of chemical reaction mechanisms in the atmosphere. Other processes (e.g., dynamics and microphysics) are simplified or neglected. Specifically, formation and loss of SOA are not included. Cloud droplets and rain were not modeled, and therefore no scavenging or wet deposition. LVOCs can dissolve and react in deliquescent (aqueous) aerosols but not in an organic apolar medium. We chose CAABA/MECCA regardless of the mentioned limitations, as this study is intended to present the chemistry scheme and demonstrate the possible influence of the update on future results. We applied the following assumptions to display the ability of the chemical mechanism to simulate experimental/observational results. Generally, we assume low-volatile organic compounds (LVOCs) as a proxy for SOA, disregarding the partitioning process. To model isoprene observation, LVOCs is defined as compounds that exhibits

an $H_s$ larger than $10^8$ M/atm (see Sect. 3.1.2). For the limonene-ozone experiment, a lower $H_s$ threshold of $4 \times 10^6$ M/atm was selected. This rule is applied to species in the gas- and the aqueous-phase, to tackle the missing condensation and new particle formation. Different thresholds were chosen, due to varying conditions affecting the importance of condensation and new particle formation and the molar mass of the major SOA products. Limonene products are more likely to condense on preexisting aerosols due to their lower saturation vapor pressure.

We intend to give an advanced evaluation of the presented mechanisms in the global model EMAC with a further extended mechanism. Due to the wide range of available submodules solving most of the above-mentioned limitations (Tost et al., 2006; Pringle et al., 2010; Tsimpidi et al., 2014; Ehrhart et al., 2018), EMAC can give a more in-depth analysis of the impact of the mechanism and advances needed in future updates.

## 2.3 Mechanism development

### 2.3.1 Overview

Table 1 lists the newly implemented and updated precursors, split into biogenic and anthropogenic origin. A major update to the chemical mechanism is introduced by updating the isoprene oxidation scheme. Additionally, the $\beta$-pinene and benzene mechanisms are revised. Limonene, IEPOX and n-alkane mechanisms were newly added. The chemistry schemes of the monoterpenes sabinene, camphene, and carene had been based on $\alpha$-pinene and were not developed for the individual compounds. They are currently excluded as they do not fulfill the standard of the implemented chemistry and will potentially be reintroduced with a more refined mechanism. The impact of these monoterpenes on LVOCs in the forward analysis is small, as depicted in fig. S9 in the supplement. In the following sections, changes to the specific mechanisms are described in detail.

### 2.3.2 Gas-phase kinetics

The nitrate radical addition to isoprene is improved based on Vereecken et al. (2021), which has been validated against chamber experiments (Carlsson et al., 2023). The former oxidation mechanism in MECCA is substituted by new pathways and products, including epoxide formation. The resulting mechanism represents a subset of the original scheme by Vereecken et al. (2021). Isoprene OH-oxidation under low $NO_x$-conditions is revised by adding the formation of epoxydiols (IEPOX) according to St. Clair et al. (2016) and dihydroxy hydroperoxy epoxides (ISOPBEPX) according to D'Ambro et al. (2017). Both compounds were previously identified as main SOA precursors (Lopez-Hilfiker et al., 2016; D'Ambro et al., 2017). Furthermore, the gas-phase oxidation of IEPOX as described by Bates et al. (2014) is included in the new mechanism. The monoterpenes $\alpha$-pinene and $\beta$-pinene are already included in the MECCA scheme with a refined mechanism. To simulate a wide range of monoterpenes, the oxidation of limonene was added to the model. The update is based on the chemical scheme of the Master Chemical Mechanism (MCM v3.3.1) by Jenkin et al. (1997) (http://mcm.york.ac.uk). If available, reaction rates are recalculated by structure-activity relationships (SARs), and low-yield pathways are excluded. Furthermore, the mechanism was refined by results of Carslaw (2013) and Vereecken and Peeters (2012). Branching ratios of the limonene-ozone mechanism

**Table 1.** List of all newly implemented and updated VOC, together with the main reactants and main mechanism sources. Compounds are divided into biogenic and anthropogenic origin.

| Precursor | Implementation type | Main reactant | Mechanism sources |
|---|---|---|---|
| **Biogenic** | | | |
| isoprene | update | $NO_3$ | Vereecken et al. (2021) |
| isoprene | update | OH | St. Clair et al. (2016), Bates et al. (2014) |
| | | | D'Ambro et al. (2017) |
| IEPOX | new | OH | Petters et al. (2021), Riedel et al. (2016) |
| limonene | new | OH / ozone / $NO_3$ | MCM, Pang et al. (2022), |
| | | | Carslaw (2013), Vereecken and Peeters (2012) |
| $\beta$-pinene | update | OH | Vereecken and Peeters (2012) |
| sabinene / carene / camphene | excluded | OH / ozone | |
| **Anthropogenic** | | | |
| benzene | update | OH | Xu et al. (2020), Wang et al. (2013) |
| pentane, hexane, | new | OH | Sivaramakrishnan and Michael (2009) |
| heptane, octane | | | Atkinson et al. (2008) |

were adapted by Pang et al. (2022). Products important to SOA are highly oxidized large hydroperoxides and ketones. Small

adjustments referring to Vereecken and Peeters (2012) were introduced for $\beta$-pinene.

Anthropogenic SOA precursors are represented by aromatics and n-alkanes. The oxidation scheme of benzene and toluene is rather detailed in the model, while higher substituted aromatics are treated in a simplified manner (Cabrera-Perez et al., 2016; Taraborrelli et al., 2021). Benzene chemistry is updated according to the results by Xu et al. (2020) who found no evidence for the epoxide channel. This was validated by previous theoretical work by Vereecken (2019). We distribute the epoxide

production into competing pathways (see Fig. S7). This adaptation yields more small, oxidized compounds like glyoxal. Thus a decrease in SOA mass in absence of cloud processing and efficient oligomer formation is expected. Similar changes for the epoxide channel were found for the OH oxidation of toluene, but at the same time the formation of alternate epoxides was shown (Wu et al., 2014; Zaytsev et al., 2019). MOM treats the oxidation of alkanes up to 4 carbons. Based on the field measurements of McDonald et al. (2018), the mechanism has been extended for n-alkanes with sizable emissions (up to n-

octane). The n-alkane mechanism is based on the work of Atkinson et al. (2008) and yields hydroperoxides, alkyl nitrates, and organic molecules containing ketone and hydroxyl groups. This mechanism is simplified and thus only covers the oxidation of specific reaction sites, and only one H-abstraction process is considered.

Experimental studies have shown partitioning of multifunctional alkyl nitrates to the aerosol phase (Perring et al., 2013). The yields of alkyl nitrates from the new $RO_2 + NO$ reactions for limonene and n-alkanes are implemented following the protocol

by Sander et al. (2019). These yields depend on the number of heavy atoms, temperature, and pressure (Arey et al., 2001; Teng

et al., 2015). Thus, the updated model is expected to generate significant amounts of SOA precursors over continental polluted regions during wintertime.

The rate constants for common reactions (e.g., OH-addition and H-abstraction) are taken from predefined functions implemented in MECCA, while rate constants for specific reactions are taken from the literature or are calculated by SAR (Kwok and Atkinson, 1995; Sander et al., 2019). In the isoprene and limonene mechanisms we consider H-shifts in peroxy radicals as described and estimated by Vereecken and Nozière (2020). In this scheme, the H-shift depends on neighboring substituents, yielding mainly highly oxygenated molecules (HOMs). New hydroperoxides either react with OH radicals by H-abstraction to reform the corresponding peroxy radical or decompose into an alkoxy and an OH radical. The decomposition step is mainly applied to molecules with neighbouring reactive groups, e.g., double bonds, which show a high reactivity towards alkoxy radicals, yielding epoxides. This kinetic scheme is key to the IEPOX and ISOPBEPX formation. In the alkane oxidation mechanism alkoxy radicals mainly undergo 1,5-H-shifts. Graphical representations of main pathways of all new reaction mechanisms are shown in Figs. S1-S8 in the supplement.

### 2.3.3  Aqueous-phase kinetics

Rate constants and branching ratios in the aqueous-phase are taken from Mouchel-Vallon et al. (2017), if available. Similar to the CLEPS 1.0 protocol, H-abstractions by hydroxyl radicals are estimated with the SAR by Monod and Doussin (2008). Only the fastest H-abstraction pathways are considered, even though the SAR provides branching ratios for the different reactive sites. The examination of all CH-bonds would require a general aqueous-phase C-H abstraction scheme in MECCA for all generated products, which is not included in the present model.

Kinetics of the IEPOX reactive uptake (acid catalyzed ring-opening) are taken from the supplementary information of Petters et al. (2021), while the branching ratios are extracted from Riedel et al. (2016). In a moderately polluted atmosphere, methylbutane-1,2,3,4-tetrol (MeBuTETROL) is the main product of aqueous IEPOX oxidation, while the corresponding organosulfate is dominant in the presence of sulfate aerosols. As an addition to the production scheme, we developed and implemented a loss mechanism based on Cope et al. (2021), which we similarly applied to ISOPBEPX. Cope et al. (2021) revealed formic acid as the main oxidation product. They proposed a mechanism, explaining the formic acid formation, invoking some reactions to proceed at unusual rates in aqueous media. Figure 1 shows our adapted and revised mechanism. Comparing the typical rate constants of the possible pathways, we redistributed the oxygen abstraction by $NO$ to the $HO_2$ elimination. The latter is nevertheless consistent with the observed formic acid yield. Similar to the mechanism by Cope et al. (2021), Fig. 1 displays the abstraction of the hydrogen atom at C4 in the mechanism. In addition, we have implemented the abstraction of the hydrogen atom at C1 after the $HO_2$ elimination yielding hydroxyacetone (not shown). Oxidation of the latter leads to the formation of the geminal diol of methyl glyoxal. The simplest geminal diol, from hydration of formaldehyde, has been shown to efficiently form formic acid by oxidation in both gas- and aqueous-phase (Franco et al., 2021). Similarly, the geminal diol of methylglyoxal yields formic acid directly but also indirectly via pyruvic acid. The related chemistry is already available in JAMOC (Rosanka et al., 2021).

As an update to earlier work, we have introduced the hydrolysis of organic nitrates in the aqueous-phase, formed from the gas-

165 phase oxidation of isoprene with OH after NO addition (Vasquez et al., 2020). Similar to the small adaptation to the isoprene $NO_3$ mechanism, we introduce the outgassing of the oligomers of glyoxal and methyl glyoxal as an update to the JAMOC mechanism. The applied $H_s$ values can be found in Table S2 in the supplement.

**Figure 1.** Simplified MeBuTETROL aqueous-phase oxidation mechanism by Cope et al. compared to the mechanism applied in MECCA. The mechanism is similarly applied for ISOPBEPX. Aqueous-phase production of formic acid is via oxidation of the geminal diol of methylglyoxal (not shown). The H-abstraction at the red hydrogen (C1; assumed to be 50%) yields additional formic acid.

## 2.4 Phase partitioning

### 2.4.1 Henry's law solubility constants

Henry's law describes the partitioning of a compound between the gas- and the aqueous-phase. In MECCA, Henry's law solubility constants $H_s$ are defined as

$$H_s = \lim_{c \to 0} c/p \tag{1}$$

where $c$ and $p$ are the equilibrium concentration and partial pressure of the compound, respectively. Most values are taken from the compilation by Sander (2023). However, for many species, especially for reaction intermediates, data are not available.

Therefore, an estimation method is required. Different approaches are available. The most widely used procedures are Henry-WIN and GROMHE. HenryWIN is developed and contributed by the US environmental protection agency (US-EPA, 2012). It estimates Henry's law solubility constants based on molecular structure descriptors (bond contribution) (Meylan and Howard, 1991). The training set consists of 345 species with additional 90 chemicals for subsequent regression and results are validated

with 72 compounds. The second Henry's law solubility constant estimation method, called GROMHE, is based on a group contribution approach. Similar to HenryWIN, the estimation is executed by multiple linear regression (Raventos-Duran et al., 2010). It uses a training set of 345 species and a validation set of 143 chemicals. Raventos-Duran et al. (2010) compared both methods and concluded that both show higher uncertainties for increasing solubilities. They revealed a large error for multifunctional compounds in HenryWIN using the GROMHE validation set. $H_s$ are overestimated for difunctional molecules, while compounds with more than two functional groups are underestimated in HenryWIN. However, running HenryWIN with the training set of GROMHE led to increased precision of the HenryWIN prediction.

Figure 2 displays the estimated Henry's law solubility constants by GROMHE and HenryWIN for the closed-shelled compounds of the newly implemented limonene mechanism. The diagram displays a good agreement between the two methods for a vast majority of the estimations, while a small subset shows a high deviation. 61% of all values lie within an order of magnitude. Larger prediction differences have a smaller impact on compounds with a generally high $H_s$, as these will partition nearly completely to the aqueous-phase in both cases. At low to medium $H_s$ the compounds which are higher estimated by GROMHE have multiple functional groups, containing mainly hydroxyl and carbonyl groups. Compounds predicted to be more soluble by HenryWIN mostly contain nitrate groups, together with further functional groups containing oxygen. A comparison to COSMOtherm values from Wang et al. (2017) does not support either of the estimation methods but lies generally in between the predictions. We use $H_s$ estimated by GROMHE. We chose GROMHE over HenryWIN because it contains a larger default training set and a better performance towards multi-functional molecules. New $H_s$ were included for compounds contained in the update. Already implemented $H_s$ were not changed.

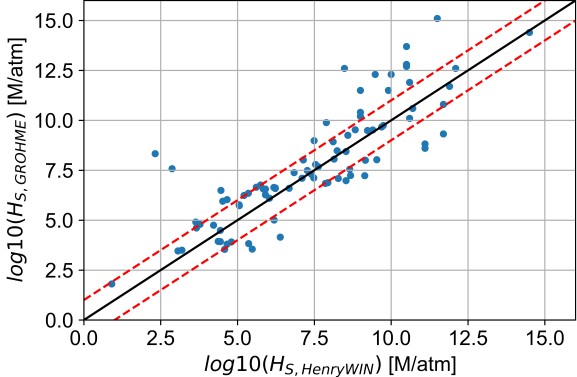

**Figure 2.** Comparison of $H_s$ estimations by GROMHE and HenryWIN for the newly implemented limonene mechanism. Both axis have a logarithmic scale. The black line represents a one-to-one comparison between both methods. The red doted lines show a deviation of one order of magnitude.

**Table 2.** B-values of group contribution and correction factors for the estimation of the temperature-dependent Henry's law solubility constants. All fragments and B-values, except for the hydroperoxy fragment, are taken from Kühne et al. (2005) and converted referring to our Henry's law solubility constant definition (multiplied by $\ln(10)$). The correction factors containing hydroxyl groups are similarly applied for hydroperoxides. The correction fragment name describes the corresponding structure in SMILES notation.

| Fragment | B-factor / [K] |
|:---:|:---:|
| basis | 1202 |
| Csb / single bond | -60 |
| Cdb / double bond | 541 |
| H | 203 |
| OH | 4145 |
| O / double bond | 2931 |
| -O- | 1966 |
| ONO2 | -811 |
| OOH | 3625 |
| **correction factors** | |
| C(=O)CO | -1538 |
| COCCO | -1538 |
| COC(OO) | -1538 |
| C(=O)O | -3009 |
| C(=O)OO | -3009 |

### 2.4.2 Temperature dependence

The van't Hoff equation describes the temperature dependence of equilibrium constants based on enthalpy. We apply it to Henry's law and define the temperature-dependence factor $B$ as

$$B = \frac{\mathrm{d}\ln H_{\mathrm{s}}}{\mathrm{d}(1/T)} = \frac{-\Delta_{\mathrm{sol}}H}{R}, \tag{2}$$

where $\Delta_{\mathrm{sol}}H$ is the enthalpy of dissolution, and $R$ is the gas constant. The substance-specific value of $B$ is often not known, in particular for large or highly-oxidized compounds. Since it can impact the partitioning considerably, it has to be estimated. Kühne et al. (2005) present an estimation approach for $B$, based on the linear combination of $B(X)$ values for predefined molecular fragments, resulting in the final value of $B$. These fragments can be single atoms (e.g., carbon in the chain) or full functional and larger groups (e.g., hydroxyl groups). Unfortunately, the hydroperoxide moiety has not been included as a molecular fragment by Kühne et al. (2005). As it plays an important role in this study, we have estimated $B(OOH)$ as the difference between $B(\text{ethane})$ and $B(\text{ethyl hydroperoxide})$. The $B(X)$ factors used in this study are shown in Tab. 2. As an example, Fig. 3 illustrates the calculation of $B(\text{methylglyoxal})$.

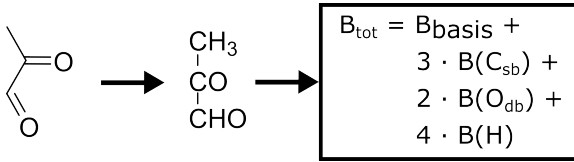

**Figure 3.** Example for the estimation of the temperature-dependence factor $B$ for Henry's law solubility constant of methylglyoxal. The subscript 'sb' denotes a single and 'db' a double bond.

The method assumes that $B(X)$ values for different fragments are additive and that the contribution of a molecular fragment is the same for every molecule it is attached to. To compensate for this, Kühne et al. (2005) introduce correction factors for specific molecular structures that can be applied to the calculated $B$ after estimation. The relevant correction factors for molecules included in the update are listed in Tab. 2. Figure 4 shows estimated values of $B$ compared to literature data. While most molecules are predicted within one order of magnitude, a tendency to overprediction can be observed for some compounds. The outliers are mainly comprised of multi-functional molecules that are affected by correction factors, implying that the corrections are not strong enough to achieve fitting values. Nevertheless, this approach reflects the trend indicated by the data from Sander (2023) and Kühne et al. (2005). This estimation approach could be adjusted using additional and stronger correction factors reflecting better the measured values. Note that B is defined differently in this work and in Kühne et al. (2005). For a direct comparison, a conversion of B is necessary (see Table 2).

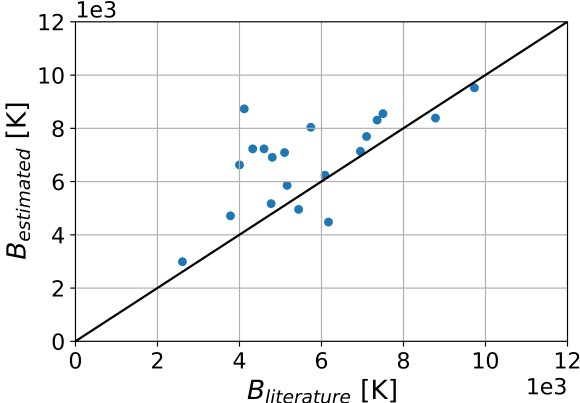

**Figure 4.** Comparison of estimated vs literature temperature dependence factor $B$ of Henry's law solubility constants. Literature data are extracted from Sander (2023) and Kühne et al. (2005).

Figure 5 shows temperature-dependent Henry's law solubility constants $H_\mathrm{s}$ for alcohols and aldehydes between 270 K and 300 K. Estimated $H_\mathrm{s}$ reasonably agree with experimental values in case of the alcohols. The aldehydes display a stronger divergence. This might be due to hydration of the aldehydes in aqueous media, or similar interactions, for which the estimation is not corrected for. Nevertheless, Fig. 5 also illustrates the need for temperature-dependent $H_\mathrm{s}$, as the constants vary over one

order of magnitude. This effect is found to be more pronounced for compounds containing multiple functional groups, as more molecular fractions with high $B$ values are applied (see Tab. S2). B values were added to the model for all $H_s$ without available temperature dependency.

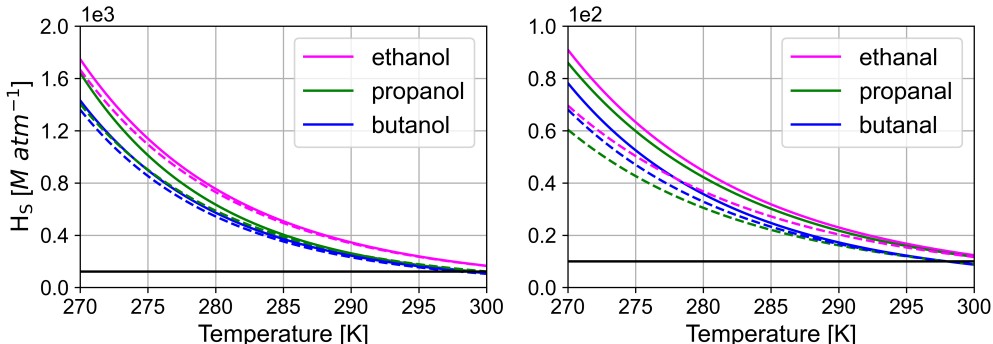

**Figure 5.** Comparison between estimated and experimental $H_s$ as a function of temperature. Solid lines denote experimental and dashed lines estimated values. The black solid line denotes $H_s$ at 298 K with no further temperature correction. Experimental data are taken from Staudinger and Roberts (2001).

## 3  Results and discussion

### 3.1  Model-model and model-observation comparison

#### 3.1.1  Model parameters/initialization

We have evaluated potential impacts of the new mechanism on model simulations by comparison to the old mechanism and in modeling IEPOX observations of the Southern Oxidant and Aerosol Study (SOAS). Model conditions are adapted to conditions during the SOAS campaign. Further details of the SOAS campaign are given elsewhere (Budisulistiorini et al., 2015; Ayres et al., 2015; Sareen et al., 2016). Table 4 displays the initial mixing ratios used in all model runs in Sect. 3.1, unless mentioned otherwise. The mixing ratio of isoprene has been fixed to 4 nmol/mol, which is the average measured during the SOAS campaign (Hu et al., 2015). For the same reason, the average aerosol salt composition during the SOAS campaign is adopted (Xu et al., 2015b) (see Tab. 3). The SOAS scenario is representative of a mildly polluted deciduous forest during summertime. All simulations are executed for a full diurnal cycle. The sensitivity runs are summarized in Tab. 5.

#### 3.1.2  BASE vs OLD run

The CAABA/MECCA box model does not simulate the production of SOA particles. Yet, a first assessment of the impact of the newly added chemistry on SOA precursors can be done by analyzing low-volatility organic compounds which can act as SOA precursors. Thus, we analyze the total (gas, aqueous and aerosol) mixing ratios of implemented LVOCs. In the scope

**Table 3.** Aerosol properties: Chemical composition, liquid water content (LWC) and particle radius. The chemical composition is averaged from Xu et al. (2015b). The LWC is taken from Nguyen et al. (2014). The average ambient temperature ($T_{ambient,SOAS}$) and particle liquid water temperature ($T_{aerosol,SOAS}$) is given to provide a better comparison between the model and observations (You et al., 2014; Nguyen et al., 2014).

| Chemical composition | | | | | | |
|---|---|---|---|---|---|---|
| $NH_4^+$ / $\frac{\mu g}{m^3}$ | $NO_3^-$ / $\frac{\mu g}{m^3}$ | $SO_4^{2-}$ / $\frac{\mu g}{m^3}$ | LWC / $\frac{\mu g}{m^3}$ | $r_{aerosol}$ / $\mu m$ | $T_{ambient,SOAS}$ / K | $T_{aerosol,SOAS}$ / K |
| 0.9 | 0.7 | 2.4 | 3.0 | 1.0 | 295.2-301.2 | 300.6 |

**Table 4.** Initial mixing ratios for all model runs. Values for NO and $NO_2$ are adapted to low, medium and high emissions in Sect. 3.1.3. The mixing ratios of $O_2$, $N_2$, $CO_2$ and isoprene ($C_5H_8$) are fixed.

| Species | mixing ratio / [nmol/mol] | Species | mixing ratio / [nmol/mol] |
|---|---|---|---|
| $H_2O_2$ | 7 | MGLYOX | 0.5 |
| $O_3$ | 25 | $C_5H_8$ | 4 |
| $O_2$ | $2.1 \times 10^8$ | PAN | 0.1 |
| $NH_3$ | 1 | $NO_3$ | $3.15 \times 10^{-3}$ |
| NO | $2 \times 10^{-2}$ | APINENE | 0.6 |
| $NO_2$ | $4 \times 10^{-2}$ | BPINENE | 0.6 |
| $HNO_3$ | $5 \times 10^{-3}$ | HCOOH | 0.35 |
| $N_2$ | $7.8 \times 10^8$ | LIMONENE | 0.6 |
| $CH_4$ | $1.86 \times 10^3$ | BENZENE | 0.1 |
| HCHO | 5 | ACETOL | 4 |
| CO | 100 | $C_5H_{12}$ | 0.4 |
| $CO_2$ | $3.5 \times 10^5$ | $C_6H_{14}$ | 0.4 |
| $CH_3CO_2H$ | 2 | $C_7H_{16}$ | 0.4 |
| $CH_3CO_3H$ | 1.5 | $C_8H_{18}$ | 0.4 |
| $CH_3OH$ | 0.5 | $CH_3OOH$ | 4 |
| HONO | $4 \times 10^{-5}$ | TOLUENE | 0.1 |

of this analysis, LVOCs are defined as VOCs fulfilling the condition $H_s > 10^8\,M/atm$, following Hodzic et al. (2014). For evaluating the impact of the new mechanism on VOCs in different molecular size ranges, the total LVOCs has been subdivided into three groups: small-LVOC (up to four carbons), medium-LVOC (five and six carbons) and large-LVOC (more than six carbons). The new mechanism has a marginal impact on the simulated small-LVOC. Hence, this class is not included in the following discussion. Figure 6 shows mixing ratios of the BASE and the OLD sensitivity run at 298 K and 278 K, respectively. Mixing ratios of key radicals during all sensitivity runs are depicted in the supplement (see Fig. S10-S14).

The medium-LVOC displays the largest impact. Before sunrise (prior 8 h), no substantial mixing ratios are depicted. After sunrise, mixing ratios increase for the BASE and the OLD run, while the BASE run displays an approximately eight times

**Table 5.** Abbreviation and description of the different setups for the sensitivity runs.

| Abbreviation | Description |
|---|---|
| BASE | base run with the updated mechanism |
| OLD | model run with CAABA/MECCA 4.5.5 |
| BASE-278K | low temperature run with the updated mechanism |
| OLD-278K | low temperature run with CAABA/MECCA 4.5.5 |
| High-$NO_x$ | model run with the updated mechanism at high $NO_x$ |
| Medium-$NO_x$ | model run with the updated mechanism at medium $NO_x$ |
| Low-$NO_x$ | model run with the updated mechanism at low $NO_x$ |

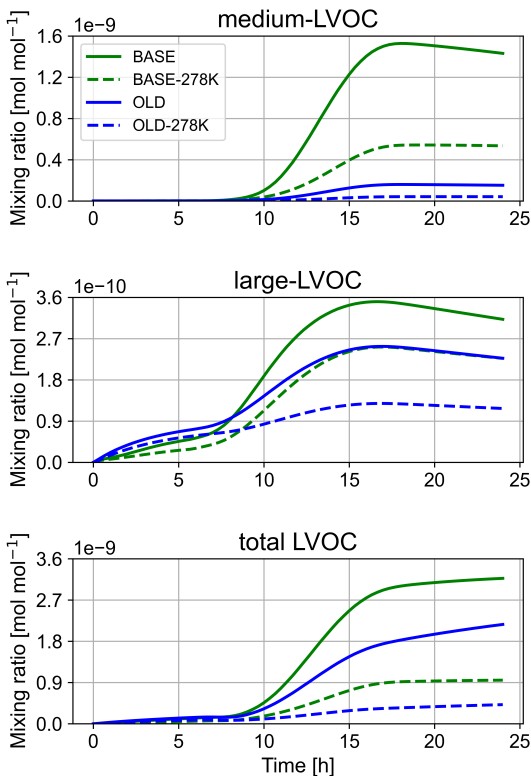

**Figure 6.** Temperature dependence of the LVOCs formation. Results from the BASE run are shown in green, while results from the OLD run are depicted in blue. The lower temperature runs BASE-278K and OLD-278K are displayed as dashed lines.

higher peak mixing ratio. This is mainly due to the formation of MeBuTETROL and organosulfates from isoprene as shown
in Fig. 7. The temperature change to 278 K lowers the medium-LVOC yield for both sensitivity runs, while the ratio between
both remains similar. For the BASE run, the concentration decrease with temperature can be explained by a decrease in IEPOX
formation in the gas-phase, due to slower gas-phase oxidation rates (see OH mixing ratios in Fig. S13), with a simultaneous

decrease of the acidity in the aqueous-phase (see Fig. S15). With the main SOA pathway of IEPOX being the acid-catalyzed ring-opening, a lower LVOC yield is expected. Nevertheless, the ratio of IEPOX between aqueous- and gas-phase increases, due to a higher partitioning coefficient at low temperatures. IEPOX products increase the aerosol yield regardless of the temperature conditions. An increased production of SOA precursors from isoprene oxidation at low NOx conditions is expected in future global model simulations (Carlton et al., 2009; Liu et al., 2016).

**Figure 7.** New aqueous-phase IEPOX scheme leading to the formation of polyols, organosulfates and oligomers.

Large-LVOC also increase during the full simulation period with the introduced update, however, a less pronounced change is predicted. Even though limonene and its oxidation pathways are newly introduced and show non-negligible LVOC yields, the exclusion of camphene, sabinene, and carene compensates partly for the additional LVOCs. For all three, a mechanism similar to $\alpha$-pinene had been assumed, due to structural similarities. The oxidation of these monoterpenes will be re-introduced as soon as new experimental/theoretical results are accessible, including a compelling mechanism for all individual compounds. Several mechanistic studies involving camphene have been published recently (Subramani et al., 2021; Afreh et al., 2021; Li et al., 2022). Only before sunrise, when oxidation by ozone is dominant, the OLD run out-competes the BASE run. This indicates a low LVOC yield in the limonene ozone mechanism for the chosen LVOC threshold. The total LVOCs show similar results as the medium-LVOC, together with a positive offset of both runs due to small-LVOC. Considering the reacted VOCs and the produced LVOCs in the BASE run we can calculate a LVOC yield to approximate the SOA yield. This analysis results in a yield of 7.3 %. With fixed isoprene concentration, it is the main LVOC contributor (approx. 90 %). Carlton et al. (2009) collected data from various studies investigating the SOA yield from isoprene and found yields up to 6 %. Taking into account that some loss processes are not implemented yet and that the aerosol formation process is not modeled, the agreement between model and experiment is reasonable.

### 3.1.3 NO$_x$ dependence

To estimate how anthropogenic NO$_x$ emissions impact LVOC (and SOA) formation, we evaluate LVOC mixing ratios under varying NO$_x$ concentrations. The main reactions of NO with peroxy radicals are O-abstraction and addition, forming alkoxy radicals and nitrates, respectively. These processes compete with the formation of hydroperoxides involving HO$_2$. As the

products from NO oxidation are generally more volatile compared to hydroperoxides, less LVOCs are formed under high NO conditions. This hypothesis is supported by Pye et al. (2010), who estimated a low contribution of "$RO_2 + NO$"-reactions to the total aerosol fraction, while the "$RO_2 + HO_2$"-reaction shows a higher fraction for all considered compounds (biogenic VOCs). However, this is not always the case for aromatic compounds. Based on chamber studies, Xu et al. (2015a) found that the SOA yield and $NO_x$ levels were correlated for toluene but anti-correlated for $m$-xylene. Figure 8 displays the model results

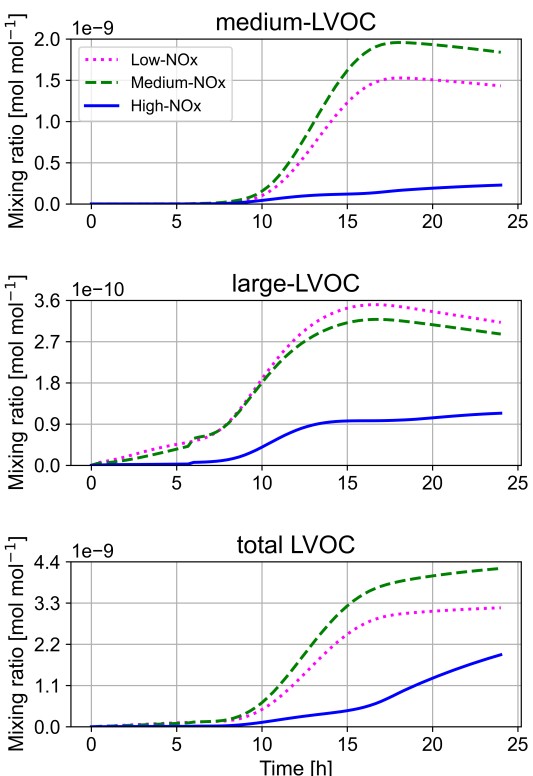

**Figure 8.** $NO_x$ dependence of the LVOC formation. Results are displayed for low-$NO_x$ (magenta, dotted line), medium-$NO_x$ (green, dashed line) and high-$NO_x$ (blue, solid line) $NO_x$ mixing ratios. For definitions of low, medium and high, see text.

of the sensitivity runs for the low-, medium-, and high-$NO_x$ scenarios. NO initial mixing ratios were set to 20 pmol/mol for low, 2 nmol/mol for medium and 20 nmol/mol for modeling highly polluted areas. $NO_2$ mixing ratios are fixed to double the amount of NO mixing ratios. The large-LVOC show the expected trend. With decreasing $NO_x$ concentration, the mixing ratios increase as a result of the higher hydroperoxide product share. Especially large-LVOC formation before sunrise from ozone

is strongly $NO_x$ dependent. The medium-LVOC show a more complex behavior. LVOC mixing ratios rise between low and medium $NO_x$ concentrations but fall off for high $NO_x$. This can be explained by the relative change of the OH concentration and the hydroperoxide yield. The OH concentration rises with increasing $NO_x$, while the hydroperoxide yield continuously decreases. IEPOX and many other LVOCs require the formation of an intermediate or product hydroperoxide group. At medium

$NO_x$ levels, the rise in the OH concentration overwhelms the decreasing hydroperoxide yield for the dominant LVOC species. This trend reverses at high $NO_x$ concentrations. Due to the comparably high total mixing ratios of the medium-LVOC bin, the total LVOC results reflect mainly the medium-LVOC.

### 3.1.4 Aqueous-phase chemistry

In the evaluation of the influence of temperature and $NO_x$ on LVOCs, the aqueous oxidation of IEPOX plays a key role and increases the complexity of the reaction system. Although the box model neglects many environmental effects and dependencies, we want to compare modeled with measured mixing ratios to see whether the model produces tracer compounds in realistic amounts. For that reason, the model setup was adjusted to the conditions during the SOAS campaign (see Sect. 3.1.1). In this campaign, Hu et al. (2015) reported measurements of MeBuTETROL together with total IEPOX-SOA concentrations (combined concentrations of MeBuTETROL, C5-triols and isoprene-derived organic sulfates) in ambient aerosol. Assuming that the main IEPOX-SOA contributors are MeBuTETROL and the corresponding organosulfates (OS), the OS concentrations can be derived by subtracting the MeBuTETROL from the total IEPOX-SOA concentration. We extracted modeled concentrations after a full model day to compare the results. Note that this comparison is only intended to show whether the model yields results of the same order of magnitude. In measured mass concentrations, only MeBuTETROL found in aerosols is taken into account and general loss pathways are accessible (e.g. deposition). The model, on the other hand, adds up all products formed (gas- and aqueous-phase), and only the chemical loss of MeBuTETROL is considered. Wet and dry deposition and volatility based condensation are neglected in the model runs, which is expected to result in an over prediction of LVOCs. Further, the modeled OH concentration exceeds the measurements ($1.2 \times 10^{-13}$ vs $5.5 \times 10^{-14}$ mol/mol, see Fig. S15 and Sanchez et al. (2018)), while the pH is higher in the model (Fig. S15) than in the SOAS aerosol (Guo et al. (2015), median pH of 1). The acid catalyzed ring-opening involving $NO_3^-$ as nucleophile, described in Eddingsaas et al. (2010), is also not considered. Thus, an over-prediction by the model is expected. Table 6 displays the mean measured and modeled MeBuTETROL, OS, and total IEPOX-SOA concentrations, with and without aqueous-phase degradation of MeBuTETROL. Modeled mass concentrations exceed the measured concentrations by an order of magnitude, while the MeBuTETROL over-prediction is 50 % higher without aqueous degradation (see Fig. 1). Considering the influence factors discussed above, this assessment lends confidence that the model for IEPOX-SOA is realistic. But it also stresses the importance of aqueous-phase degradation pathways in models. Non-negligible loss processes are still missing in the MECCA chemical scheme. The aqueous processing of all hydroperoxides, alkyl nitrates, and -sulfates is not implemented, resulting in an overestimation of those species. Additionally, there is no SAR method dealing with the reactivity of epoxides towards OH radicals in aqueous media. Reaction rates have to be estimated by treating epoxides like hydroxyl groups, not considering the enthalpy gained by prompt ring-opening. On the other hand, organic coatings on aqueous aerosols and the aerosol phase state may limit the reactive uptake of IEPOX and thus the production of MeBuTETROL and OS (Zhang et al., 2018; Octaviani et al., 2021).

In addition to the LVOC concentrations, a change in the O/C ratio might occur as a result of the chemistry update. Especially, because LVOCs from aqueous oxidation is added, which commonly possesses a higher O/C ratio (Lim et al., 2010). The main LVOC components, MeBuTETROL and OS, exhibit a ratio of 1.0 and 1.6, respectively. Thus, an increase in the O/C ratio

**Table 6.** Comparison between measured and modeled MeBuTETROL, OS and total IEPOX-SOA mass concentrations. Measured values are taken from Hu et al. (2015). The concentrations were investigated over multiple days, but a mean concentration of $0.3\ \frac{\mu g}{m^3}$ is assumed in the scope of this comparison. Modeled mixing ratios are extracted after a full modeled day.

| Compound | Measured / $\frac{\mu g}{m^3}$ | Modeled / $\frac{\mu g}{m^3}$ | Modeled no MeBuTETROL degrad. / $\frac{\mu g}{m^3}$ |
|---|---|---|---|
| | Hu et al. (2015) | This work | |
| MeBuTETROL | 0.3 | 4.4 | 6.6 |
| OS | 0.6 | 5.3 | 4.4 |
| total IEPOX-SOA | 0.9 | 9.7 | 11.0 |

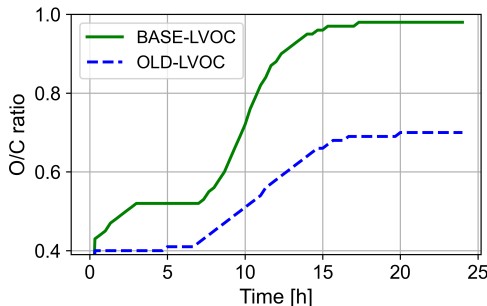

**Figure 9.** O/C ratio of summed up LVOCs for the BASE and OLD run.

is expected. Figure 9 displays the O/C ratio of the aqueous LVOCs for the BASE and OLD run over the simulation time. After a full modeled day, an increase of 40 % is found, with a ratio of 0.98 and 0.70, for BASE and OLD, respectively. The updated model is in good agreement with the SOAS campaign average O/C ratio of 0.91 (Massoli et al., 2018). More accurate degradation pathways of MeBuTETROL and OS are expected to further improve the O/C ratio.

In addition to the already described aqSOA formation sources, there are further known, but missing processes not covered by the mechanism. Formation of nitroaromatics from the aqueous-phase oxidation of anthropogenic VOCs is an important additional source of aqSOA. For instance, it has been shown that 4 % to 20 % of total SOA in Europe is aqSOA originating from residential wood burning (Gilardoni et al., 2016). Accounting for this source of aqSOA requires the development of an explicit mechanism for heterogeneous reactions of $N_2O_5$ where chloride and phenols compete with water for the addition to $NO_2^+$ (Heal et al., 2007; Ryder et al., 2015; Hoffmann et al., 2018; Staudt et al., 2019). A simplified scheme for the nitration of phenol has recently been implemented into MECCA by Soni et al. (2023). Moreover, consideration of nitroaromatics during nighttime oxidation of furans from biomass burning will further increase the model predictions of SOA mass (Joo et al., 2019; Al Ali et al., 2022).

### 3.2 Model-chamber comparison

#### 3.2.1 Experimental setup and considerations

Model results are compared to data from two experiments carried out in the SAPHIR (Simulation of Atmospheric PHoto-chemistry in a large Reaction chamber) chamber for further validation of the chemical mechanism. More information about the SAPHIR chamber can be found elsewhere (Rohrer et al., 2005; Karl et al., 2006; Schlosser et al., 2007). The newly implemented limonene mechanism is tested by simulating an experiment on the ozonolysis of limonene in the dark with subsequent aerosol aging by $NO_3$ (Gkatzelis et al., 2018). Thereby, the performance of the modified MCM scheme is assessed (see Sect. 2.3.2). The potential impact of the new mechanism on SOA is investigated by looking into the organic mass and the aerosol O/C ratio. The O/C ratio measured by an aerosol mass spectrometer (AMS) is used in an already corrected form, as presented in Canagaratna et al. (2015). The full isoprene-NO3 mechanism by Vereecken et al. (2021) was previously compared to results of chamber experiments in SAPHIR (Vereecken et al., 2021). Since the mechanism included in this work is a subset of this scheme, we decided to compare it with the same chamber experiment to investigate, whether using the subset gives similar results. Thus, we focus on gas-phase reactions and products. Alkyl nitrate (AN) and peroxy radical ($RO_2$) concentrations from model simulations based on the old and new isoprene-$NO_3$ mechanisms were compared to each other. $RO_2$ concentrations are additionally compared to measured data. For $RO_2$ measurements, some adjustments were applied to the model results. $RO_2$ radicals are not directly measured by the laser-induced fluorescence (LIF) instrument used in the campaign. To detect $RO_2$, they are first converted to alkoxy radicals in the reaction with NO. Formed alkoxy radicals are then expected to undergo H-abstration by $O_2$ yielding $HO_2$, which is measured by the LIF (Fuchs et al., 2008). Thus, measurement results depend on the $HO_2$-yield from the given alkoxy radical. For some alkoxy radicals like the methyl alkoxy radical, this yield is close to 100 % (Novelli et al., 2021; Vereecken et al., 2021). However, Novelli et al. (2021) investigated competitive reaction pathways for alkoxy radicals containing a nitrate group. In some cases, they found that decomposition ($NO_2$-elimination) and isomerization reactions can compete with the $HO_2$ production. This complicates model-experiment comparisons. To fit model results better to the $RO_2$ detectable by the LIF instrument, Vereecken et al. derived the subset for individual Isoprene-$NO_3$ products that yields $HO_2$ (see SI of Vereecken et al. (2021)). Similar to Vereecken et al. (2021) we use these values to calculate the detectable $RO_2$ from the total $RO_2$. In contrast to the mechanism by Vereecken et al. not all compounds with stereochemistry are included with all isomers in the subset scheme, but are treated as lumped species. For those species, the correction for detectability is done as if the lumped species consist of 50 % of each isomer. Due to the strong dependency of the results on the $NO_3$ and $HO_2$ concentrations, model concentrations were constrained to measured data. Isoprene injections were adjusted to reproduce the observed increase in the concentration measurements of the VOCUS instrument (see Brownwood et al. (2021)). VOCUS data were corrected by a factor of 0.7 to account for the measurement's dependency on water.

Chamber injections were modeled as the increase of the injected chemical to the measured peak values within one timestep. In Table 7 all injections are listed. For the limonene and isoprene experiment dilution effects are modeled using a rate of $4.8\%\mathrm{h}^{-1}$ and $5.6\%\mathrm{h}^{-1}$, respectively. Dilution rates are calculated from the mean total flow during the experiments. For the limonene experiment, this rate is applied to gas and aerosol species. The high volume-to-surface ratio of the SAPHIR chamber minimizes

the wall losses and are not considered as a first approximation. Especially, particle wall loss is discussed in Sect. 3.2.2. The aerosol liquid water content (ALWC) is estimated based on AMS measurements of the high resolution $H_2O$ signal, which represents an approximate upper limit. This signal is influenced by interference of organics. Further information about the experiments is shown in Section 5 of the SI.

**Table 7.** Injections used in modeling of the chamber experiments. Model injections refer to an instantaneous change of the given mixing ratio after the previous timestep.

| Limonene + Ozone | | | | |
|---|---|---|---|---|
| Injection No. | Limonene [ppb] | Ozone [ppb] | NO [ppb] | time since start [min] |
| 1 | 23.6 | 140 | - | 0 |
| 2 | - | - | 30 | 450 |
| Isoprene + NO$_3$ | | | | |
| Injection No. | Isoprene [ppb] | Ozone [ppb] | NO$_2$ [ppb] | time since start [min] |
| 1 | - | 105 | 24 | 0 |
| 2 | 5.1 | - | - | 5 |
| 3 | 8.0 | 108 | 23 | 117 |
| 4 | 7.0 | 102 | 25 | 202 |
| 5 | - | 111 | 23 | 360 |

### 3.2.2 Limonene + Ozone

The modeled and measured organic mass concentration (OM) is displayed in Fig. 10. Both modeled and measured mass concentrations increase rapidly at the start of the experiment. The experimental OM decreases after one hour, while the model still predicts an increase. This is due to missing loss processes in the model as only the dilution of gas- and aerosol-phase species is considered, although species are also lost to the chamber walls. Schmitt (2018) has shown, that particle losses to chamber walls in the SAPHIR chamber can be higher than the dilution. Figure S18 shows results for the organic mass with

simplified aerosol wall loss, with a loss rate similar to Schmitt (2018). This test simulation indicates, that a larger portion of the over-prediction is due to missing particle loss. The nucleation and condensation of pure organics is also not modeled (see Sect. 2.2). Additionally, in the model, large compounds (> $C_4$) in the aerosol phase are currently not further oxidized (except isoprene and methylglyoxal products) (Rosanka et al., 2021), resulting in an insufficient chemical loss of organics. This missing oxidation is also reflected in the aerosol O/C ratio. In the experiment, the measured O/C ratio is in the range

between 0.56-0.66, while the model predicts values between 0.4-0.51 with a similar trend (see Fig. S16). The nitrogen oxide injection after 7.5 h leads to an OM increase in both simulation and experiment. As a result of the overall higher OM and thus in total more oxidizable compounds, the model results show a steeper OM increase after the NO injection.

The experimental data was modeled without constraining the radical concentrations to experimental values. OM and O/C

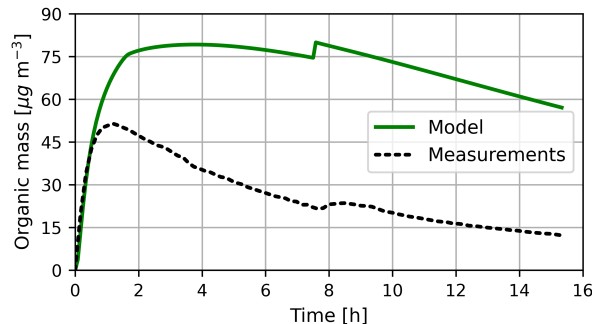

**Figure 10.** Mass concentration of OM in model and experiment. Modeled OM represents all species with an $H_s$ of larger than $4 \times 10^6$ M/atm in either gas- or aqueous-phase. This simplification is applied to account for missing processes in the model. After 7.5 h NO is injected into the chamber.

ratio values are strongly connected to the amount of limonene consumed by the different radicals (OH, ozone). As the OH concentration were not measured, the modeled OH concentration cannot be validated with experimental results. Thus, model results might be affected by incorrect radical concentrations. Figure S17 displays the OH reactivity in the model and the experiment, implying an underestimation of OH in the model.

### 3.2.3   Isoprene + NO$_3$

The modeled and measured organic peroxide radical ($RO_2$) mixing ratios are displayed in Fig. 11. As the new isoprene-NO$_3$ mechanism is an update to the previously used mechanism (in contrast to limonene), experimental results are compared to the old and new mechanism representations. The detectable $RO_2$ is only determined for the new mechanism, as the $HO_2$-yields are unknown for the $RO_2$ formed in the old scheme (see Sect 3.2.1).

All model simulations over-predict $RO_2$ concentrations, especially shortly after the VOC injections. Measured $RO_2$ level trends after the initial peaks show reasonable agreement between the experiment and detectable $RO_2$ of the new mechanism. Especially during times, when the $RO_2$ concentration decreases, the detectable $RO_2$ is in good agreement with the experimental results. The old mechanism shows an offset during times when the $RO_2$ concentration decreases, indicating the formation of a sufficiently stable $RO_2$. The $RO_2$ results are comparable to the simulation by Vereecken et al. (2021). Simulation results are influenced by how model variables are constrained to measured data. In Vereecken et al. (2021) constrained radical levels are fitted to experimental data, while in CAABA/MECCA concentrations are set to the exact value measured. Thus, radical ($HO_2$ and $NO_3$) concentrations in Vereecken et al. (2021) are higher, close to the injection than in the present study.

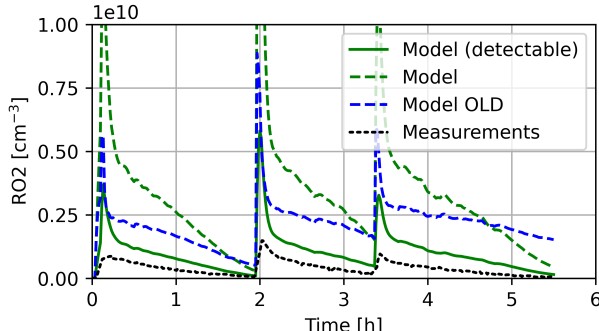

**Figure 11.** $RO_2$ mixing ratios in model and experiment. Model results marked with "detectable" are modified by $HO_x$ formation efficiencies by Vereecken et al. (2021).

Figure 12 displays the AN mixing ratio as a function of isoprene consumed by $NO_3$ calculated by rate constants and concentrations from the model. Mixing ratios predicted by the new mechanism are 10-20% lower than results from the old scheme. Although a higher variety of AN products are accessible, in the new scheme, also more loss processes are considered. Especially compounds, that are only lost to OH radicals in the old mechanism, like 1-Hydroxy-2-propanone nitrate (NOA),

keep the AN concentration at high levels in the old run. Throughout the isoprene-$NO_3$ campaign, an average of $108 \pm 15$ % of ANs was derived which is in reasonable agreement with the model simulations with the new mechanism (84-93 % AN yield). AN yields of the old mechanism are even closer to the measurements (98-116% AN yield). The slight under-prediction of the new model is likely due to the decomposition of organic nitrates forming small oxidized compounds and $NO_2$ in the degradation process. The faster AN loss processes are a potential artifact of the missing reactions from the original scheme.

Nevertheless, various products in the new scheme are expected to increase the modeled SOA yield in future global simulations. Among them are functionalized epoxides and hydroperoxy nitrates.

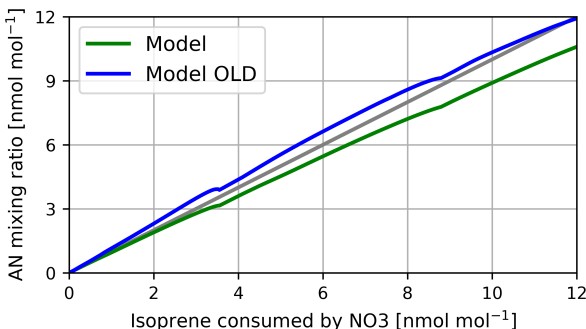

**Figure 12.** AN mixing ratios against isoprene consumed by $NO_3$. The gray line displays the one-to-one line.

## 4 Summary and Outlook

We updated the chemical scheme of MECCA to improve the production of SOA precursors in the model. In addition, we investigated the gas-/aqueous-phase partitioning of the new species formed. To assess the update, we investigated the production of LVOCs at different physical conditions and initial concentrations in a box model. As expected, we find an increase in the total mixing ratio of LVOCs and also the change in LVOCs of different sizes is more pronounced. Results display a rising production of medium- and large-sized LVOCs, while the number of small LVOCs stays constant. This is due to the new implementation of exclusively medium to large precursors. By changing the temperature from 298 K to 278 K, the model predicts a decrease in LVOC production, while an increase in SOA yield is expected. This can be attributed to the simplified way SOA is accounted for. The $NO_x$-dependence shows more complex patterns of change. In general, the LVOC yield decreases with rising $NO_x$, but for medium-LVOC, an increase at moderate $NO_x$ is found. This is due to the aqueous-phase IEPOX oxidation scheme. The modeled O/C ratio confirms the aerosol measurements of the SOAS campaign. Model simulations of chamber experiments generally aligned with experimental results, especially considering the model limitations influencing the limonene-ozone experiment. Overall, we find that the model responds differently to newly added aqueous- and gas-phase reactions. Gas-phase processes are well modeled by the new mechanism. To also better reflect the processing of VOCs and LVOCs in clouds and aerosols, more aqueous-phase reactions, also for the small and large compounds are needed. Furthermore, the implementation of recent findings concerning the peroxy radical reactivity might lead to additional reaction pathways or altered branching ratios Schervish and Donahue (2021); Mayorga et al. (2022). H-shifts of peroxy radicals are only considered in the new limonene mechanism and might lead to an increased formation of HOMs in the model if included for all compounds (Wu et al., 2021). Similarly, the treatment of peroxy radical dimerization was found to be more important than previously estimated (Schervish and Donahue, 2020). In future work, we will evaluate the impact of the new mechanism on SOA precursors with explicit multiphase kinetics in deliquescent aerosols and cloud droplets in the global model EMAC (Rosanka et al., 2023). To investigate the dependence of results on the partitioning scheme and the used Henry's law solubility constants at the global scale, sensitivity runs will be executed with varying partitioning scheme. Further, the introduction of more SOA/LVOC loss processes to the model has shown to be important, as already pointed out by Hodzic et al. (2016). A general aqueous-phase degradation scheme of organic nitrates and hydroperoxides would further refine SOA processes.

*Code availability.* The updated MECCA model code is available as a community model published under the GNU General Public License (https://www.gnu.org/copyleft/gpl.html). The model code can be found in the Supplement (DOI:10.5281/zenodo.7944174) and in the code repository at https://gitlab.com/RolfSander/caaba-mecca. In addition to the complete code, a list of chemical reactions including rate constants and references (meccanism.pdf) and a user manual (caaba_mecca_manual.pdf) are available in the manual directory of the supplement. A list of all Henry's law and accommodation constants (chemprop.pdf) is available in the tools/chemprop directory. For further information and updates, the CAABA/MECCA web page at http://www.mecca.messy-interface.org can be consulted.

*Author contributions.* FW and DT designed the study and developed the chemical mechanism. The latter was reviewed by all co-authors and implemented into MECCA by FW and RS. FW performed the estimation of the Henry's law coefficients. The manuscript was prepared by
450 FW and reviewed by all co-authors. The measurements were carried out and analyzed by CC, AN, HF, RT and TH.

*Competing interests.* Some authors are members of the editorial board of GMD. The peer-review process was guided by an independent editor, and the authors have also no other competing interests to declare.

*Acknowledgements.* The authors gratefully acknowledge the computing time granted through JARA on the supercomputer JURECA (Jülich Supercomputing Centre, 2021) at Forschungszentrum Jülich.

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
