# Peer review of "Development of a multiphase chemical mechanism to improve secondary organic aerosol formation in CAABA/MECCA (version 4.5.6-rc.1)"

_Geoscientific Model Development, 2023_

## Author Comment (AC3)

**Reply on Referee #1**

This paper describes the 3D-atmospheric model MESSy for a comprehensive evaluation of the impact of aqueous-phase chemistry on SOA in a CAABA/MECCA box model platform. The authors need to clarify the fundamental assumption and the limitation of the model. This manuscript may be suitable to be published in ACP after major revision by considering the com-

5 ments as listed below.

10

Thank you very much for the helpful comments, text and figure corrections!

1. The authors should clarify the criteria of the SOA model. It is unclear that the SOA model of this paper is subjective to aqueous droplet or any aerosol including SOA aerosol without aqueous phase. For the organic and inorganic mix, it is possible for the aerosol to be phase separable into organic phase and inorganic phase or form homogeneous phase. Without the assumption and the suitable criteria of the model to a specific aerosol conditions, it is difficult to understand the usage of model.

- CAABA/MECCA does not contain a specific submodule which describes SOA. In the manuscript, SOA is accounted as low volatile organic compounds (LVOC) that exhibit an Henry's law solubility constant larger than  $10^8$  M/atm (see Sect. 3.2) and are present in gas or aqueous phase. These LVOC can only dissolve and react in deliquescent aerosols but not in a organic apo-
- 15 lar medium. The partitioning of LVOC to the latter will be considered in the global model, where MECCA is coupled to a SOA submodule based on the VBS (Pozzer et al., 2022). This box model study is intended to demonstrate the possible influence of the update of the chemical mechanism. Our results are relevant for conditions with relatively high humidity and temperature where glass-transition and phase-separation are not expected. Nevertheless, a global model study assessing the impacts of the mechanism update on the predictions of SOA distribution is planned. To clarify this more generally we extended the abstract
- 20 with the following addition "The extensions in CAABA/MECCA will be ported to the 3D-atmospheric model MESSy for a comprehensive evaluation of the impact of aqueous-phase chemistry on SOA at a global scale in a follow up study.".

2. There is no demonstration of model against laboratory data or simulation in regional scale. It is hard to know whether the simulation of the SOA model is suitable to apply to ambient air.

25 We see the point of Referee #1. However, our manuscript is meant as a *model description* paper as defined by the GMD journal. We plan an evaluation of the updated model against laboratory/observational data, but this will be submitted as a separate paper (see above).

3. In addition to IEPOX, there are other chemical species that can form oligomers via acid-catalyzed reactions. The authors 30 needs to clarify limitation in the model.

We agree with Referee #1 that there are additional species able to form oligomers via acid-catalyzed reactions. Examples are the oligomerization of glyoxal, which has been shown to yield more particular matter under acid catalysis or further epoxides other than IEPOX (Jang and Kamens, 2001; Lim et al., 2010). Oligomers from glyoxal and methyl glyoxal are already implemented in our mechanism (Rosanka et al., 2021). To the previous version of the scheme we add the acid catalyzed reactive

uptake and oligomerization of IEPOX as this is known to be among the most sources of SOA. 35

4. Line 9 (Abstract): Only up to C8 but it is characterized as "long-chain alkanes". Probably not appropriate to say C8 and below are long-chain alkanes considering they do not form any SOA.

- We agree, this is stated incorrectly in the manuscript. We changed the line to "n-alkanes (5 8 C atoms)" and deleted "longchained" in following lines. We also agree that C5-C8 n-alkanes have a comparably low SOA yield, but there are several 40 impact factors and limitations to consider. Alkanes with a high SOA yield are predominantly high molecular weight alkanes, as the yield increases with chain length. On the other hand, the implementation of these compounds would require a much more sophisticated mechanism with many more reactions to account for the SOA formation. Including all of them would highly complicate and slow down global multiphase chemistry simulations, due to a strong increase in memory and computational demand
- for the integration of the very stiff chemical ODE. Compared to cyclic alkanes,  $\beta$ -scission of the alkoxy radicals from linear 45 alkanes is more likely to yield smaller and more volatile compounds. However, this is generally the case for highly elevated NOx concentrations under which experiments have been conducted, e.g. Lim and Ziemann (2009). The corresponding SOA yields for alkanes are then often used in air quality models, e.g. Pye and Pouliot (2012). However, under low-NOx conditions it

is known that the RO2 from oxidation of large alkanes react mostly with HO2 yielding compounds with hydroxy, hydroperoxy
and carbonyl moieties (Srivastava et al., 2022). The latter are of lower volatility compared to the carbonyl compounds under high-NOx. Additionally, emission data of various alkanes are scarce. We decided to incorporate n-alkanes up to C8, to include alkane SOA at least partly in simulations. Especially for octane, measurements show a moderate SOA yield (in comparison to other alkanes) and observations show high emission rates (see supplementary material of McDonald et al. (2018)).

55 **5.** Line 12 (Abstract): "aqueous phase" has no hyphen here but is hyphenated in the previous sentence (Line 11). We adjusted "aqueous phase" to "aqueous-phase" in the entire manuscript.

**6.** Line 14 (Abstract): "LVOC is..." should be replaced with "LVOC are..." We changed this in the revised manuscript.

60

**7.** Line 33. Please provide a more detailed explanation on the impact of aqueous-phase oxidation products on the vertical distribution of SOA in the atmosphere would enhance the understanding.

Generally, aqueous- and gas-phase oxidation takes place at different timescales (Seinfeld and Pandis, 2016) and aqueous reactions depend on available cloud or aerosol water, aqueous phase properties (pH) and dissolved pollutants, while both reaction

- 65 types depend on gas phase pollutants. This can impact model simulations in different ways, depending on the chosen scenario. Carlton et al. included in-cloud secondary organic aerosol formation (processing of glyoxal and methylglyoxal) in the CMAQ model and found a general increase of organic PM mass, but an especially strong increase at high altitudes (Carlton et al., 2008). Models have significant issues in reproducing the vertical profile of measured concentrations of organic aerosols, e.g. Pai et al. (2020). Also the global model of which MECCA is part of is no exception (Pozzer et al., 2022). Therefore, we expect
- 70 lower model biases when our updated chemical mechanism is used in the global model EMAC. We adjusted the paragraph in the manuscript as follows "Additionally, products from aqueous-phase oxidation are formed at different time-scales than from gas-phase oxidation. Combined with aqueous- and gas-phase properties (pollutants, pH), this impacts the vertical distribution of SOA in the atmosphere depending on the simulation setup (Seinfeld and Pandis, 2016; Hodzic et al., 2016).".
- 75 **8.** Line 90: "... small oxidized compounds like " Should be "... small, oxidized compounds like glyoxal." We changed this in the revised manuscript.

**9.** Figure 1: Unclear what happens in the abstraction of the red H step. It seems like some compound must be being released from the mechanism other than the ones shown. I can see maybe H2O from the OH and one O atom from the radical but how is the red H involved?

80 is the red H involved?

Fig 1. displays the abstraction of the blue hydrogen and subsequent reactions. The red hydrogen can be abstracted similarly, and the color is meant to identify the hydrogen as alternative route (described in the caption). We agree that the missing depiction of the released molecule can lead to confusion, thus it was added to the figure together with an indicator that the red hydrogen poses an alternative route.

85

**10.** Henry's solubility. In aqueous aerosol, there will be electrolytic inorganic species which can influence the solubility of organic species. The salt composition and humidity can influence organics' Henry's constants. The authors need to clarify the impact of salts on gas-particle partitioning of organic species.

We agree with Referee #1, that "salting in" and "salting out" can influence the Henry's law solubility constants. However,
these effects usually do not change the solubility by more than an order of magnitude (Yu and Yu, 2013) and cannot explain the up to 106 times larger Henry's law coefficients for small organics derived from field measurements (Nah et al., 2018; Shen et al., 2018). Nevertheless, even potential large salting-in effects have been shown to be insufficient for explaining high concentrations of organic aerosols in winter haze events in China (Gkatzelis et al., 2021). Additionally, reaction rate constants can depend on the ionic strength of the solvent (Mekic and Gligorovski, 2021). These non-linear effects are not accounted for in the

95 model, due to their complexity and unknown influence parameters (Sander, 2015). The authors are not aware of any influence of humidity on Henry's law solubility constants. We extended Sect. 2.1 with "Non-linear effects like "salting in" and "salting out" influence Henry's law solubility constants. The model does not account for these effects, due to the implementation complexity and insufficient data availability (Sander, 2015).".

**100 11.** Section 2.3.1: Phase partitioning is simulated through Henry's law constants. Can this model handle dry conditions without an aqueous phase? It seems like this would be necessary in a global SOA model.

In CAABA/MECCA the aqueous phase can be deactivated to simulate dry condition, but the model does not include an additional aerosol submodule. Thus, compounds would remain in the gas phase. In the global model MESSy, which shares the same chemical mechanism (MECCA), refined aerosol microphysics and various kinds of uptake processes are included and can be used under different conditions (Pringle et al., 2010; Tsimpidi et al., 2014; Pozzer et al., 2022).

12. Please check the legend in Fig. 5. Legend for aldehydes is missing.

The aldehydes do have a separate legend in the right-hand plot of Fig. 5. They are divided in a similar color scheme as chosen for the alcohols.

110

105

**13.** Line 215. Please check figure number. It should be Fig. S17, not Fig. S11. The Figure reference has been changed in the revised manuscript.

14. Line 216 - 218. Is this because the concentration of the BASE run is high regardless of temperature conditions (Fig. 6)?
While you have provided a reference, further explanation about why the increase in the SOA precursor from isoprene is expected in the global model would be needed.

This is correct. As an increase in LVOC is found at varying conditions compared to the previous chemical mechanism and this change can be attributed to the IEPOX mechanism, a total increase of isoprene derived SOA precursors is expected in the global model. In the current mechanism, aqueous processing of isoprene products is not considered, even though IEPOX

120 reactive uptake was found to be the most dominant SOA contributor (from isoprene). Thus, an increase of the total isoprene SOA is expected in global model runs, especially in low NOx regions (see Marais et al. (2016) for IEPOX contribution).

**15.** Line 224. The previous sentence stated that the oxidation by ozone and NO3 is dominant before sunrise. What is the effect of the limonene NO3 mechanism? Table 1 mentioned that the main reactant of d-limonene is ozone. Please briefly describe the effect of NO3 on d-limonene.

and total LVOC are barely influenced. We will include these reactions in the next mechanism update! The discussion of the

125 effect of NO3 on d-limonene. We have to admit that the presented update does not contain limonene NO3 reactions. To display the influence of limonene NO3 anyhow, we have extended the mechanism with missing reactions for test purposes. The resulting large-LVOC do not change significantly at nighttime, but the large-LVOC at the end of the full modeled day increases by 4.8%. Medium-LVOC

130 present manuscript is scarcely impacted by the missing reactions.

**16.** Line 254. What are the factors that the box model neglects and what are the potential effects? With respect to aerosols the box model neglects dry and wet deposition, a complex scheme describing aerosol micro-physics, condensation of gases regarding their volatility (solely Henry's law is taken into account) and a general scavenging scheme.

135 This limits the possible scenarios that can be simulated by the box model. Potential effects on the presented results are the over prediction of liquid phase species due to missing deposition. For dry scenarios, the partitioning scheme additionally over predicts liquid phase species with high Henry's law coefficients. In sect. 3.4 we added "Wet and dry deposition and volatility based condensation are neglected in the model runs, which is expected to result in an over prediction of LVOC.".

140 17. Line 256. Please provide the detail conditions of SOAS campaign applied in this study (e.g., temperature, humidity, and pollutant concentrations).The SOAS conditions or combined for the concentrations (events a concentration from SOAS) and the isogram.

The SOAS conditions are applied for the aerosol salt concentrations (average salt concentration from SOAS) and the isoprene mixing ratio, to achieve a general estimation regarding the IEPOX SOA products. We extended table 3 and added a link to the detailed description in Sect. 3.1 to the text.

145

**18.** Line 265 – 266. Please check figure number.

The Figure reference has been changed in the revised manuscript.

19. Line 271 – 272. I wonder if the model setup was based on the average value of the entire SOAS campaign or a specific case

- 150 of SOAS campaign. If there are special cases showing high/low NOx or high/low isoprene conditions during the campaign, box model can be performed for those cases. If the results show a good agreement with measurements in those cases, it will show the good quality of the box model under various conditions. If the final goal of this box model is to improve prediction of SOA concentration by applying it to the global model, it is necessary to show that the box model can produce reasonable results under more various conditions than the paper presents.
- 155 The model setup is based on average values from the SOAS campaign. Here it has to be noted that the box model approach was chosen, because of the much simpler test environment in comparison to the global model. We agree with Referee #1 that a more detailed review of the impact of the new chemistry under various conditions is necessary. But these simulations are planned to be executed in the global model, where more aerosol submodules are accessible (less limitations in the model) and a more in detail analysis is possible. In this manuscript, our main aim is to present the new chemistry update and display that
- 160 the use of the new scheme likely will result in a more realistic formation of LVOC, with the major compounds being IEPOX products.

**20.** Line 297 - 298. The model was performed under various conditions and the results were presented. It would be beneficial to emphasize the significance of the results for each specific condition. If this box model is applied to the global model to predict

165 the SOA concentration in global scale, please suggest expected outcomes and their potential role considering the variations in NOx and temperature conditions.
The SOA viald is expected to be inversely preparticulate the NO.

The SOA yield is expected to be inversely proportional to the  $NO_x$  concentration. This is due to the higher volatility of reaction products from  $NO_x$  chemistry compared to products from competing reactions (e.g., hydroperoxide formation) for most dominant VOCs. The temperature influences multiple factors at the same time. Reactions are generally slower at lower

170 temperatures and saturation concentrations in the liquid phase are increased. On the other hand, condensation of low volatile compounds on available aerosols is favored and more condensed aqueous phase is available at low temperatures. Generally, it is expected that SOA formation is anti-proportional to the temperature. We specified the simulated scenario by adding "This scenario is representative of a mildly polluted deciduous forest during summertime." to sect. 3.1 of the revised manuscript.

**References**

215

- 175 Carlton, A. G., Turpin, B. J., Altieri, K. E., Seitzinger, S. P., Mathur, R., Roselle, S. J., and Weber, R. J.: CMAQ model performance enhanced when in-cloud secondary organic aerosol is included: comparisons of organic carbon predictions with measurements, Environmental science & technology, 42, 8798–8802, 2008.
  - Gkatzelis, G. I., Papanastasiou, D. K., Karydis, V. A., Hohaus, T., Liu, Y., Schmitt, S. H., Schlag, P., Fuchs, H., Novelli, A., Chen, Q., Cheng, X., Broch, S., Dong, H., Holland, F., Li, X., Liu, Y., Ma, X., Reimer, D., Rohrer, F., Shao, M., Tan, Z., Taraborrelli, D., Tillmann, R., Wang,
- H., Wang, Y., Wu, Y., Wu, Z., Zeng, L., Zheng, J., Hu, M., Lu, K., Hofzumahaus, A., Zhang, Y., Wahner, A., and Kiendler-Scharr, A.: Uptake of Water-soluble Gas-phase Oxidation Products Drives Organic Particulate Pollution in Beijing, Geophysical Research Letters, 48, e2020GL091351, https://doi.org/10.1029/2020GL091351, \_eprint: https://onlinelibrary.wiley.com/doi/pdf/10.1029/2020GL091351, 2021.
- Hodzic, A., Kasibhatla, P. S., Jo, D. S., Cappa, C. D., Jimenez, J. L., Madronich, S., and Park, R. J.: Rethinking the global secondary
   organic aerosol (SOA) budget: stronger production, faster removal, shorter lifetime, Atmospheric Chemistry and Physics, 16, 7917–7941, https://doi.org/10.5194/acp-16-7917-2016, 2016.
  - Jang, M. and Kamens, R. M.: Atmospheric secondary aerosol formation by heterogeneous reactions of aldehydes in the presence of a sulfuric acid aerosol catalyst, Environmental Science & Technology, 35, 4758–4766, 2001.
- Lim, Y., Tan, Y., Perri, M., Seitzinger, S., and Turpin, B.: Aqueous chemistry and its role in secondary organic aerosol (SOA) formation,
   Atmospheric Chemistry and Physics, 10, 10521–10539, https://doi.org/10.5194/acp-10-10521-2010, 2010.
- Lim, Y. B. and Ziemann, P. J.: Effects of Molecular Structure on Aerosol Yields from OH Radical-Initiated Reactions of Linear, Branched, and Cyclic Alkanes in the Presence of NOx, Environmental Science & Technology, 43, 2328–2334, https://doi.org/10.1021/es803389s, publisher: American Chemical Society, 2009.
- Marais, E. A., Jacob, D. J., Jimenez, J. L., Campuzano-Jost, P., Day, D. A., Hu, W., Krechmer, J., Zhu, L., Kim, P. S., Miller, C. C., et al.:
   Aqueous-phase mechanism for secondary organic aerosol formation from isoprene: application to the southeast United States and cobenefit of SO 2 emission controls, Atmospheric Chemistry and Physics, 16, 1603–1618, 2016.
  - McDonald, B. C., De Gouw, J. A., Gilman, J. B., Jathar, S. H., Akherati, A., Cappa, C. D., Jimenez, J. L., Lee-Taylor, J., Hayes, P. L., McKeen, S. A., et al.: Volatile chemical products emerging as largest petrochemical source of urban organic emissions, Science, 359, 760–764, 2018.
- 200 Mekic, M. and Gligorovski, S.: Ionic strength effects on heterogeneous and multiphase chemistry: Clouds versus aerosol particles, Atmospheric Environment, 244, 117 911, https://doi.org/10.1016/j.atmosenv.2020.117911, 2021.
  - Nah, T., Guo, H., Sullivan, A. P., Chen, Y., Tanner, D. J., Nenes, A., Russell, A., Ng, N. L., Huey, L. G., and Weber, R. J.: Characterization of aerosol composition, aerosol acidity, and organic acid partitioning at an agriculturally intensive rural southeastern US site, Atmospheric Chemistry and Physics, 18, 11471–11491, https://doi.org/10.5194/acp-18-11471-2018, publisher: Copernicus GmbH, 2018.
- 205 Pai, S. J., Heald, C. L., Pierce, J. R., Farina, S. C., Marais, E. A., Jimenez, J. L., Campuzano-Jost, P., Nault, B. A., Middlebrook, A. M., Coe, H., Shilling, J. E., Bahreini, R., Dingle, J. H., and Vu, K.: An evaluation of global organic aerosol schemes using airborne observations, Atmospheric Chemistry and Physics, 20, 2637–2665, https://doi.org/10.5194/acp-20-2637-2020, publisher: Copernicus GmbH, 2020.
- Pozzer, A., Reifenberg, S. F., Kumar, V., Franco, B., Kohl, M., Taraborrelli, D., Gromov, S., Ehrhart, S., Jöckel, P., Sander, R., Fall, V., Rosanka, S., Karydis, V., Akritidis, D., Emmerichs, T., Crippa, M., Guizzardi, D., Kaiser, J. W., Clarisse, L., Kiendler-Scharr, A., Tost,
- 210 H., and Tsimpidi, A.: Simulation of organics in the atmosphere: evaluation of EMACv2.54 with the Mainz Organic Mechanism (MOM) coupled to the ORACLE (v1.0) submodel, Geoscientific Model Development, 15, 2673–2710, https://doi.org/10.5194/gmd-15-2673-2022, publisher: Copernicus GmbH, 2022.
  - Pringle, K. J., Tost, H., Message, S., Steil, B., Giannadaki, D., Nenes, A., Fountoukis, C., Stier, P., Vignati, E., and Lelieveld, J.: Description and evaluation of GMXe: a new aerosol submodel for global simulations (v1), Geoscientific Model Development, 3, 391–412, https://doi.org/10.5194/gmd-3-391-2010, publisher: Copernicus GmbH, 2010.
- Pye, H. O. T. and Pouliot, G. A.: Modeling the Role of Alkanes, Polycyclic Aromatic Hydrocarbons, and Their Oligomers in Secondary Organic Aerosol Formation, Environmental Science & Technology, 46, 6041–6047, https://doi.org/10.1021/es300409w, publisher: American Chemical Society, 2012.
- Rosanka, S., Sander, R., Wahner, A., and Taraborrelli, D.: Oxidation of low-molecular-weight organic compounds in cloud droplets: develop ment of the Jülich Aqueous-phase Mechanism of Organic Chemistry (JAMOC) in CAABA/MECCA (version 4.5. 0), Geoscientific Model Development, 14, 4103–4115, https://doi.org/10.5194/gmd-2020-337, 2021.

Sander, R.: Compilation of Henry's law constants (version 4.0) for water as solvent, Atmospheric Chemistry and Physics, 15, 4399–4981, https://doi.org/10.5194/acp-15-4399-2015, 2015.

Seinfeld, J. H. and Pandis, S. N.: Atmospheric chemistry and physics: from air pollution to climate change, John Wiley & Sons, 2016.

- 225 Shen, H., Chen, Z., Li, H., Qian, X., Qin, X., and Shi, W.: Gas-Particle Partitioning of Carbonyl Compounds in the Ambient Atmosphere, Environmental Science & Technology, 52, 10997–11006, https://doi.org/10.1021/acs.est.8b01882, publisher: American Chemical Society, 2018.
- Srivastava, D., Vu, T. V., Tong, S., Shi, Z., and Harrison, R. M.: Formation of secondary organic aerosols from anthropogenic precursors in laboratory studies, npj Climate and Atmospheric Science, 5, 1–30, https://doi.org/10.1038/s41612-022-00238-6, number: 1 Publisher:
   Nature Publishing Group, 2022.
- Tsimpidi, A. P., Karydis, V. A., Pozzer, A., Pandis, S. N., and Lelieveld, J.: ORACLE (v1.0): module to simulate the organic aerosol composition and evolution in the atmosphere, Geoscientific Model Development, 7, 3153–3172, https://doi.org/10.5194/gmd-7-3153-2014, publisher: Copernicus GmbH, 2014.

235

Yu, X. and Yu, R.: Setschenow Constant Prediction Based on the IEF-PCM Calculations, Industrial & Engineering Chemistry Research, 52, 11 182–11 188, https://doi.org/10.1021/ie400001u, publisher: American Chemical Society, 2013.

---

## Author Comment (AC4)

**Reply on Referee #2**

The manuscript by Wieser et al. presents an updated multiphase chemical mechanism in the CAABA/MECCA model, designed to improve predictions of secondary organic aerosol (SOA), particularly via aqueous oxidation pathways. The manuscript was submitted as a development and technical paper, which describe technical updates leading to model improvements, including

5 new parameterizations. Such papers are expected to include a significant amount of evaluation. The manuscript largely fits this description and is appropriate for GMD. However, the manuscript is lacking clarity and specificity in several key areas, which are noted below. It is recommended that these areas be addressed before the manuscript is further considered for publication in GMD.

Thank you for your helpful comments and editorial corrections!

**10**

**Major scientific comments:**

**1.** Processes and phases included in the base version of CAABA/MECCA. The language regarding SOA formation is sometimes unclear in the context of whether this model only includes SOA formation via aqueous uptake (aqSOA) or whether gas-particle partitioning to a mostly organic phase is also included. This needs to be better clarified and considered in the

- 15 discussion of the model updates and results. It is particularly confusing in 3.2, where it is unclear what is being evaluated. The authors state that analysis of the LVOCs is useful as an assessment on potential SOA formation, but then also go on to say they analyzed the total gas, aqueous, and aerosol mixing ratios. The results in 3.2 and 3.3 seem to focus only on LVOC production, to understand how much aqSOA forms, the authors also need to couple that with aqueous solubility and reactive uptake. Additionally, if the compounds are of sufficiently low volatility, there will be competition for partitioning to a mostly organic
- 20 phase if present. On pp. 15-16, it is not clear what O/C ratios are being compared. Literature reported O/C ratios for SOA are typically based on bulk composition of the condensed phase (organic and/or aqueous depending on analytical technique). It seems that those values are being compared with the O/C ratio for all LVOCs, independent of phase. These are not the same thing.

The model does not account for SOA formation from either aqueous uptake or volatility-based partitioning, as submodules for

25 these processes are not available in the box model we have used here. How competitive these processes are will be tested at a second stage by means of global model simulations. This box model study is meant to give a first assessment of the impact of the improved and extended oxidation mechanism.

The O/C ratio is calculated from all compounds categorized as LVOC in the gas and the aqueous phase. We agree that it is more appropriate to calculate the O/C ratio for the aqueous phase only. In warm and humid environments, like the one during

30 the SOAS campaign, we expect aerosol water to be important for the phase partitioning of LVOC. Therefore, we think that a comparison with model results is meaningful. We adapted the O/C calculation accordingly.

**2.** Activity coefficients. The manuscript has no discussion of activity coefficients, or effective Henry's law constants (i.e., salting in or salting out). This is a major omission. The effective Henry's law constants of organic compounds in the aqueous

35 particles will be composition dependent and may deviate significantly from the Henry's law constant for a pure water. It is particularly important for the compounds with moderate solubility. This has been widely discussed in the literature from both experimental and theoretical perspectives.

We agree with Referee #2, that "salting in" and "salting out" can influence Henry's law solubility constants. These non-linear effects could be easily accounted for in the model. However, they are not yet considered because of the lack of Sechenov

- 40 constants (Sander, 2015). On the other hand, the enhancement of effective Henry's law coefficients relative to the intrinsic ones is strongly affected by aqueous-phase chemical reactions as well. For instance, it is known that the hydration of carbonyl compounds like formaldehyde and glyoxal increases the Henry's law coefficients by 2-3 orders of magnitude. This major effect is explicitly accounted for by our kinetic model. We extended Sect. 2.1 by the mentioned model limitations "Non-linear effects like "salting in" and "salting out" influence Henry's law solubility constants. The model does not account for these effects, due
- 45 to the insufficient data availability (Sander, 2015)."

**3.** Mechanism details. While the mechanism details can be obtained from the code and other external documents, it is suggested that the SI be expanded to more clearly define the differences between the original and updated mechanism and any changes

- 50 that were made to published data (e.g., reaction rate constants, branching ratios, etc.) as implemented in the new mechanism. The text is very minimal and non-specific on these aspects, which reduces the clarity and reproducibility of this work. To provide some examples: 1) On p. 3 line 76, the authors discuss redistributing the product yields for isoprene + NO3. How are these redistributed? Does the affect any prior performance evaluations? 2) The authors describe that the new chemical mechanism for limonene was based on MCM, and on p. 4 line 83 state "low-yield reaction pathways are excluded". What was the cutoff yield
- value? How was this determined? 3) Similarly, the authors describe implementation of the new n-alkane mechanism based on Atkinson et al. (2008), and on line 96 state that the "mechanism is simplified" to only cover oxidation at specific sites and only one H-abstraction process. Which sites? How was this determined? Does this affect prior performance evaluation? What is the implication of these choices in the context of this model application? Line 115-What is considered fast? What is the rate (s-1) cutoff value? These are only some examples, but such detail is important for all new mechanisms and updates presented in this manuscript, especially if modified from published literature.
- We agree with Referee #2, that the manuscript lacks some information regarding the choices made in the mechanism creation. We excluded too in-depth explanations from the text to ensure readability and convenience of the text. To give an improved overview of the mechanism and important assumptions/simplifications, we have extended the "Reaction kinetics and mechanisms" chapter in the SI by dividing it into additional sub-chapters containing further information.
- 65

70

**4.** The evaluation should focus on how the mechanism updates affect the performance of CAABA/MECCA as has been previously evaluated, and using experimental data where available. There is too much general comparison with one-off published literature results, which are inconclusive in the context of model performance, and too much speculation of how the mechanism will improve global model performance. Evaluation is also needed for the new Henry's law constants and their dependence on temperature, as compared with the base model.

- A comparison to earlier evaluations of CAABA/MECCA concerning the content of the mechanism update is hardly possible. This is due to the fact that mostly radical and VOC mixing ratios were analyzed in previous assessments, rather than the stable products (Sander et al., 2011, 2019; Taraborrelli et al., 2012; Nölscher et al., 2014; Hens et al., 2014; Mallik et al., 2018). The comparison between the OLD and the BASE run is intended to display how and in which size range the model is impacted
- 75 by the update. With secondary organic aerosols being the main interest of further investigations, we decided to restrict this to LVOC, as these are a proxy for SOA. Additionally, Figs. S10-S14 display the change in key radical mixing ratios (NO, NO2, NO3, OH, and O3)(previously Figs. S10-S16). We improved the diagrams to give a more comprehensive overview of the differences between the runs. Ultimately, this work is a new approach to comparing changes to the mechanism by looking into specific properties of products. This can be used as the basis for future evaluations.
- 80 Henry's law coefficients are only newly included for species introduced with the mechanism update. For species in the base model, the temperature dependency was added, if missing. Therefore, a direct comparison to the base model would not lead to conclusive results. We indicated this in Sec. 2.3 in the revised manuscript. Further replacements and additions together with a complete assessment of the newly added constants are planned for future work.

**85 Minor scientific comments:**

1. The authors note that limonene was added to simulate a wide range of monoterpenes. It would be useful to know what other monoterpenes are included. Limonene is known as having one of the highest SOA yields under most (if not all) reaction conditions. Thus including limonene to expand the range makes sense only if monoterpenes with low yields are well represented. Related to comment #2 above, what small adjustments based on Vereecken and Peeters were introduced for beta-pinene? The

90 authors note lack of a compelling chemical mechanism for monoterpenes including camphene. Several camphene mechanism papers have been published recently including: Li et al., 2022 ACP; Subramani et al., 2021 Chemosphere; Afreh et al., 2021 ACP, and Mehra et al., 2020 ACP.

 $\alpha$ - and  $\beta$ -pinene are implemented in the model with a refined mechanism. The carene, camphene, and sabinene schemes were implemented based on  $\alpha$ - and  $\beta$ -pinene (similar products and kinetics were assumed). We agree with Referee # 2, that there

95 are sufficient papers on the camphene oxidation mechanism, to create a refined scheme. Nevertheless, we decided to update the mechanisms of the listed monoterpenes, if compelling information for all of them is available. To clarify this in the text, we

added/modified "The oxidation of these monoterpenes will be re-introduced as soon as new experimental/theoretical results are accessible, including a compelling mechanism for all individual compounds. Several mechanistic studies involving camphene have been published recently (Li et al., 2022; Subramani et al., 2021; Afreh et al., 2021)." to Sec.3.2.

100

110

**2.** Line 100: The statement about significant amounts of SOA precursors is unexpected in the methods and also unsupported. We agree with Referee #2 and excluded the statement.

**3.** Line 163: What is the rationale for choosing GROHME over HenryWIN? This is not made clear in the discussion.

105 GROMHE has a larger training set and shows better performance towards multi-functional molecules (Raventos-Duran et al., 2010). We added the statement "We chose GROMHE over HenryWIN because it contains a larger default training set and a better performance towards multi-functional molecules."

**4.** Line 200: Does this mean that the model does not include new particle formation? Or does it mean that gas-particle partitioning to an organic phase is not considered (see also major scientific comments #1)?

- The model does not include particle formation. The effects on both processes will be tested in the global model. Aerosol nucleation is treated in the MESSy submodels NAN and IONs (Ehrhart et al., 2018). SOA formation is specifically treated by ORACLE (Tsimpidi et al., 2014).
- 115 5. Line 217-218: The suggestion of increased isoprene SOA precursors is unclear and unsupported. Increase because of what? In a global model you also have the temperature dependence of emissions and possibly deposition, so I'm not sure how the box model sensitivities are being translated to expectations for a global model. Similar comment on lines 226-227: There is not strong support for the logic connecting the box model observations (particularly when many monoterpene mechanisms are not included) to expected global model results.
- 120 We agree with Referee #2, that box model results do not intrinsically implicate similar changes in the global model. However, it was already demonstrated in experiments and models that the consideration of IEPOX and consequent products enhances SOA precursors and yield from isoprene oxidation (Carlton et al., 2009; Budisulistiorini et al., 2017). Thus, the increased LVOC yield in the box model together with literature results raises the expectation of an increase of SOA precursors, at least under low-NOx conditions.
- 125 We agree with Referee #2's comment on lines 226-227. The exclusion of the monoterpenes complicates the assessment of whether the total SOA yield is increased with the updated scheme. We excluded the statement.

**Editorial comments:**

We have reordered the figures in the supplementary information and corrected the typo.

---

## Author Response (AR2)

**Author's response:**

"Dear authors, thank you for submitting a revised manuscript. In a very critical second-round review it was suggested to reject the manuscript due to the lack of scientific evaluation of the model developments (especially against laboratory data), a lack of a clear and concise presentation, and an insufficient translation of the responses made by the authors to the reviewer's report into changes of text within the manuscript. It is standing policy at GMD to generally reject manuscripts rated poor in at least one category I do mostly concur with the reviewer's points of view.

The amount of comparison against observations is indeed very limited. A number of recent field studies (e.g. ATom, ACCLIP), also under conditions where strong SOA production is expected like a wildfire (e.g., BBFLUX, WE-CAN, FIREX), as well as chamber experiments (e.g., CLOUD, AIDA) are readily available to test the model developments under a much broader range of conditions, which would lend much more credibility to the usefulness of the developments.

Responding in detail to points raised by a reviewer but not translating these changes into substantial changes within the manuscript text is a common fallacy - one that leads an improper representation of the peer-review process within the final manuscript. The final manuscript must reflect the agreed-upon state after discussion between reviewers and authors. All the (highly useful) scientific content within the review comments and author responses would be lost to the reader (of the published paper) otherwise.

In summary, I do think the developments shown in the manuscript merit publication, but considering the justified criticism by the reviewer and given standing policy at GMD to generally reject manuscripts rated poor in at least one category, I would like to ask the authors for major revisions consisting of (1) a proper inclusion of responses to the reviewer into changes within the manuscript text and (2) adding more evaluation against observations before we can proceed."

We agree with the reviewers and the editor, that some reviewer responses were not properly represented in the revised manuscript. The implementation of a new section on model limitations and the additional evaluation that was possible within the revision time frame is hopefully sufficient to satisfy the major issues raised by the reviewers.

**Major revision referring to (1)**

Major comments were reevaluated and the most significant modifications to the manuscript are included as follows.

- We added Section 2.2 "model limitations" to the manuscript. With some more additions to the "Specification" section this includes all responses to questions on model capabilities and limitations (Referee #1, comments: 1, 11 and 16; Referee #2, Major comment: 1, Minor comment: 4).

- In the "Specification" section we included further statements on the influence of non-linear effects on Henry's law coefficients (Referee #1, comment: 10; Referee #2, Major comment: 2).

- We extended the mechanism details given in the supplement by adding the reactions displayed in the figure in kpp format. The figures were also revised to enhance comprehensiveness (Referee #2, Major comment: 3).

**Major revision referring to (2)**

Comparison to measured data was added to the manuscript as follows.

- For further evaluation against measured data, we added Section 3.2 to the manuscript (Referee #1, comment: 2). This Section presents the setup and comparison of model simulations to chamber experiments. The limonene-ozone and isoprene-NO3 mechanisms are evaluated. With the assessment of the IEPOX chemistry in Section 3.1, this covers a wide range of the newly implemented mechanism.

- Additional information on the experimental setup and the model-experiment comparison is given in Section 5 of the supplement.